# Callus growth kinetics and accumulation of secondary metabolites of *Bletilla striata* Rchb. f. using a callus suspension culture

**Yinchi Pan**[1,2]☯, **Lin Li**[1]☯, **Shiji Xiao**[3], **Zhongjie Chen**[1], **Surendra Sarsaiya**[4], **Shebo Zhang**[1], **Yanni ShangGuan**[1], **Houbo Liu**[1], **Delin Xu**[1]*

**1** Department of Cell Biology, Zunyi Medical University, Zunyi, Guizhou, P. R. China, **2** Department of Science and Education, Zhejiang Putuo Hospital, Zhoushan, Zhejiang, P. R. China, **3** Department of Pharmacy, Zunyi Medical University, Zunyi, Guizhou, P. R. China, **4** School of Pharmacy Chemistry, Zunyi Medical University, Zunyi, Guizhou, P. R. China

☯ These authors contributed equally to this work.
* xudelin2000@163.com

**Data Availability Statement:** All relevant data are within the paper and its Supporting Information files.

## Abstract

*Bletilla striata* is an endangered traditional Chinese medicinal plant with multiple uses and a slow regeneration rate of its germplasm resources. To evaluate the callus growth kinetics and accumulation of secondary metabolites (SMs), a callus suspension culture was proven to be a valuable approach for acquiring high yields of medicinal compounds. An effective callus suspension culture for obtaining *B. striata* callus growth and its SMs was achieved with the *in vitro* induction of calluses from *B. striata* seeds. The callus growth kinetics and accumulation of SMs were analyzed using a mathematical model. The resulting callus growth kinetic model revealed that the growth curves of *B. striata* suspension-cultured calluses were sigmoidal, indicating changes in the growth of the suspension-cultured calluses. Improved Murashige and Skoog callus growth medium was the most favorable medium for *B. striata* callus formation, with the highest callus growth occurring during the stationary phase of the cultivation period. Callus growth acceleration started after 7 days and thereafter gradually decreased until day 24 of the cultivation period and reached its highest at day 36 period in both the dry weight and fresh weight analyses. The coelonin concentration peaked during the exponential growth stage and decreased afterward during the stationary stage of the callus suspension culture. The maximum content of coelonin (approximately 0.3323 mg/g callus dry weight) was observed on the 18th day of the cultivation cycle, while dactylorhin A and militarine reached the highest concentrations at day 24, and *p*-hydroxybenzyl alcohol at day 39. This investigation also laid a foundation for a multimathematical model to better describe the accumulation variation of SMs. The production of SMs showed great specificity during callus growth and development. This research provided a well-organized way to increase the accumulation and production of SMs during the scaled-up biosynthesis of calluses in *B. striata* callus suspension cultures.

**Funding:** This research was financially supported by the National Natural Science Foundation of China (31560079, 31560102), the Scientific Project of Guizhou Province (QKH-ZY[2013]3002, QKH-LH [2014]7549, [2017]5733-001), Talents Promotion Project of Zunyi Medical University(JC2018-2-5 (1)), the PhD Science Foundation of Zunyi Medical University (F-809), and the Talent Growth Project of Guizhou Education Department (KY[2017]194). The funders did not play any role in the study design, data collection and analysis, decision to publish, or preparation of the manuscript.

**Competing interests:** The authors have declared that no competing interests exist.

## Introduction

*Bletilla striata* is a perennial herb of Orchidaceae and is an important traditional Chinese herbal medicine recorded in the pharmacopoeia of all previous dynasties. It is sweet and slightly cooling, with prominent effects of healing muscles and stopping bleeding. Hence, it is ideal for treating traumatic bleeding[1]. At present, due to the limitation of traditional artificial breeding methods, it is difficult for the production of *B. striata* tubers and their medicinal active ingredients to meet the market demand[2]. *B. striata* has become an endangered class with decreasing wild plant populations due to recent over-utilization. For the sustainable progress and comprehensive consumption of *B. striata*, it is essential to recognize its callus growth dynamics with the accumulation of SMs. However, until now, limited approaches have been developed on these perspectives. In addition, the quality and yield of medicinal substances have been limited because of the deprivation of cultivated variations over the course of long-period cultivation. Hence, it is necessary to improve the medicinal properties of *B. striata* with upright quality, greater yield, and optimal production of important secondary metabolites for sustainable production of this medicinal resource.

Plant callus suspension culture technology can achieve artificial control to provide the optimal conditions for callus growth and for the differentiation and accumulation of SMs; therefore, we can efficiently promote callus proliferation and directionally induce the synthesis and accumulation of secondary metabolites[3]. This technology has become the most promising biosynthetic method for producing secondary metabolites from plant calluses. Callus suspension culture systems of various medicinal plants have been established at national and global levels, but studies on callus suspension cultures and the detection of secondary metabolites of *B. striata* have rarely been reported[4,5]. In the early stages, our research group isolated, purified and identified a variety of secondary metabolites from the tubers of *B. stri*ata[6,7]. However, it is still unknown whether these secondary metabolites are also present in *B. striata* suspension-cultured calluses.

Based on the baseline research of our group, *B. striata* seeds have successfully been used to induce callus and establish a rapid propagation system[8]. Based on the induction and proliferation of *B. striata* callus, in this paper, we used an induced loose callus as the initial material to establish an optimized callus suspension culture system and generated the growth curve. Changes in the accumulation of four major SMs, *p*-hydroxybenzyl alcohol, dactylorhin A, militarine and coelonin, were also evaluated. These studies laid a foundation for the further development of *B. striata* callus suspension culture bioreactors, as well as genetic improvement, regulation of callus proliferation, SM production, and improvement of the efficiency of producing pharmaceutically important ingredients by using *B. striata*.

## Materials and methods

### Experimental materials

The *Bletilla striata* seeds used in this study were purchased from Menghe Ran, a farmer planting medicinal herbs in Zheng'an County (28°56′N, 107°43′E), Guizhou Province of China, on October 20[th] of 2015 with the price of 10 RMB per capsule. Then the seeds were only tissue cultured in labs of Chinese Medical Herb Research Group on main campus of Zunyi Medical University locating in Xinpu District, Zunyi City, Guizhou Province of China. All of the experiments conducted on the seeds, seedlings and plants by the research group were thoroughly comply by the requirements on plant researches of common ethics and the rules of the University.

The *B. striata* plants for harvesting capsules were first collected by Ran's family from Zheng'an areas many years ago. The farmer who saled materials to the research group completely agreed all of the researches on the materials, including landraces' capsule seeds, plants and tubers bought from him and hoping the researchers can get improvement for helping his business.

The seeds were germinated into seedlings by tissue culture approach for harvesting the research materials. Four months later, we planted the seedlings in fields for the next year's flowering. We collected undissected seed pods 15 weeks after pollination and used the mature seeds to induce loose callus. The seeds were tissue cultured in the labs of the Chinese Medical Herb Research Group on the main campus of Zunyi Medical University located in Xinpu District, Zunyi City, Guizhou Province of China. All of the experiments conducted on the seeds, seedlings and plants by the research group thoroughly complied with the requirements on plant research of common ethics and the rules of the university. The callus was then inoculated on Murashige and Skoog (MS) medium supplemented with 1.0 mg/L 6-benzylaminopurine (6-BA), 3.0 mg/L 2,4-dichlorophenoxyacetic acid (2,4-D), 30 g/L sucrose, and 7 g/L agar powder and subcultured in the dark at 25˚C. After 2 generations of approximately 30 days, callus with good growth, loose texture and uniform growth was selected as the explant of the liquid suspension culture.

HPLC grade methanol for Burdick & Jackson ACS/HPLC was procured from Honeywell, USA, and the standards dactylorhin A (CAS: 256459-34-4), militarine (CAS: 58139-23-4), coelonin (CAS: 82344-82-9) were purchased from ChemFaces Corp. The standard *p*-hydroxybenzyl alcohol (CAS: 623-05-2) was purchased from Chengdu Ruifensi Biotechnology Corp., Ltd., China. The purity of all reference substances was greater than 98%.

## Instruments

A BL-100A autoclave and a GZX-9146MBE electric drying oven were purchased from Shanghai Boxun Medical Bio Instrument Co., Ltd., China; a CJHS ultraclean workbench from Tianjin Taisite Instrument Co., Ltd., China; an Agilent 1260 HPLC, DAD UV detector, and ChemStation chromatography workstation from Agilent Company, United States; and a BT125D analytical balance 1/100,000 from Sartorius Company, Germany.

## Construction of the callus suspension culture system

The previously obtained loose, tender yellow callus of *B. striata* was inoculated into 35 mL liquid medium (MS, 1.0 mg/L 6-BA, 3.0 mg/L 2,4-D and 30 g/L sucrose) in a 100 mL flask with pH 5.8 at 1.0 g callus per flask (fresh weight). After inoculation, the flask was placed on a rotary shaker with a shaking speed of 120 r•min$^{-1}$ at 25˚C under dark culture conditions.

## Determination of dry weight, fresh weight and the growth curve

Based on the culture conditions stated above, the fresh and dry weights were measured at 3-day intervals after inoculation up to 45 days; this was considered to be one cycle. During the study, the flask containing the suspension callus fluid on the shaker was removed and then shaken and filtered until no droplets formed. The fresh weight was obtained and then dried in an oven at 50˚C to a constant weight. The dry weight was also measured. From each sample point, three samples were collected and measured three times. The fresh and dry weights were recorded, and the growth curve of the calluses was plotted with the culture time as the abscissa and the fresh and dry weights as the ordinate.

## Callus growth curve modeling method

According to the data of the fresh and dry weights of suspension calluses, the growth curve was plotted and analyzed by Logistic, Boltzmann and DoseResp with Origin 9.1 software. The best fitted model was determined by the coefficient of determination ($R^2$), and the F value was analyzed by ANOVA[9]. The selected function model was used to detect the acceleration rate of callus growth to analyze the proliferation of calluses.

## Extraction and detection of secondary metabolites

**Preparation of test solution and chromatographic conditions.**   According to the above sampling method outlined the callus of different growth stages was used to detect the content of four target secondary metabolites. The sampling was repeated 3 times at 3-day intervals for each detection point. The callus was filtered, dried, crushed, and passed through a sieve (Φ 200×50 mm). Approximately 0.20 g of *B. striata* callus was accurately weighed, mixed with 100 mL of 70% methanol water for 2 hours in a condensation reflux extraction, and centrifuged to obtain an extract liquid. The recovered extract liquid was dried under reduced pressure, dissolved in an appropriate amount of 70% methanol in water, and then transferred to a 5 mL volumetric flask for use as a test solution. HPLC detection conditions were as follows: column, Dubhe $C_{18}$ (250 mm × 4.6 mm, 5 μm); mobile phase, methanol (A), ultrapure water (B); flow rate, 0.8 mL/min; column temperature, 25°C; detection wavelength, 225 nm[10]; and injection volume, 20 μL. The mobile phase gradient program was as follows: 0–10 min, 20%-25% A; 10–25 min, 25%-50% A; 25–35 min, 50%-50% A; 35–50 min, 50%-100% A; and 50–60 min, 100%-100% A.

**Study on the linear relationship of the standard curve.**   A suitable concentration of *p*-hydroxybenzyl alcohol, dactylorhin A, militarine, and coelonin were prepared in a standard solution and diluted with methanol to prepare a series of mass concentrations (0.25, 0.2, 0.15, 0.1, 0.05, and 0.025 mg/mL for *p*-hydroxybenzyl alcohol, 1.0, 0.8, 0.6, 0.4, 0.2, and 0.1 mg/mL for militarine, and 0.1, 0.05, 0.025, 0.01, 0.005, and 0.0025 mg/mL for coelonin). The detection was performed under standard chromatographic conditions as described in the section of test solution preparation.

**Validation of the HPLC method.**   Precision experiment. The standard solutions were precisely made as former described, and 20 μL was injected for HPLC analysis. The standard solutions were detected under the chromatographic conditions outlined in section of test solution preparation and repeated 6 times.

Stability experiment. A sample solution (as described in section of test solution preparation) was injected at 0, 3, 6, 9 and 12 hours after preparation. Peak areas of *p*-hydroxybenzyl alcohol, dactylorhin A, militarine and coelonin were recorded.

Repeatability of experiment. Five *B. striata* callus clusters were collected, each of which weighed 0.20 g. The samples were prepared according to the above method described. All samples were injected under the same chromatographic conditions. The injection volume was 20 μL. The peak areas were recorded, and the mass fractions of *p*-hydroxybenzyl alcohol, dactylorhin A, militarine and coelonin were calculated. The relative standard deviation (RSD) was calculated according to the formula.

$$\mathrm{RSD}(\%) = STDEV/AVERAGE \times 100$$

$$SD = \sqrt{\frac{\Sigma(\mathrm{xi} - \mathrm{x})^2}{n-1}}$$

Sample recovery analysis. Nine *B. striata* suspension-cultured callus clusters with known secondary metabolites were accurately weighed, each of which weighed 0.02 g, and were precisely spiked with three reference solutions with low, medium and high mass concentrations (80%, 100%, 120% of the original sample, respectively). There were 3 reference substances at each mass concentration. The recovery rate and RSD of each component were calculated.

## Modeling method for cumulative curve of secondary metabolites

The data of the cumulative number of secondary metabolites of *B. striata* suspension calluses were fitted by a variety of function models under the "Nonlinear Curve Fit" option in Origin 9.1 software, and the best fitting model was determined by the coefficient of determination ($R^2$).

## Results

### Growth curve of suspension-cultured calluses

The growth curve of the suspension culture calluses was plotted with the fresh and dry weights as the indicators. As shown in Fig 1, the growth curves of the two different growth indicators were basically the same, and both were sigmoidal in shape. The morphology of the fresh cultured callus was almost the same (S1 Fig).

### Growth curve and kinetic characteristics

**Functional model of growth curve.**  After obtaining the callus weights of dry and wet calluses from the suspension-cultured samples during the 45-day culture period (S1 Table), mathematical models were fitted and analyzed for changes in fresh and dry weights of the suspension-cultured calluses. The results are shown in Table 1.

As shown in Table 1, the coefficients of determination of the Logistic, Boltzmann and DoseResp fitting equations for the fresh weight versus culturing time were 0.9817, 0.9231, and 0.9780, respectively. The coefficients of determination of the three fitting equations for the dry weight versus culturing time were 0.9761, 0.8878 and 0.9453, respectively. According to the fitting results, the Logistic model was more suitable for describing the growth of biomass in the suspension-cultured calluses of *B. striata*. After a further significance test, the F values of the Logistic equation (721.56294 and 613.94951) were also higher than those of the Boltzmann equation and the DoseResp equation, indicating that the curve of the Logistic equation is more consistent with the experimental data. The Logistic equation was further used to plot the curve of the fresh and dry weight growth of the suspension-cultured calluses (Fig 2).

The results showed that the whole culture cycle could be divided into six stages: the lag stage, exponential stage, linear stage, deceleration stage, stationary stage and recession stage. Among them, the lag stage was determined to be from day 0 to 6. During this period, the fresh and dry weight of suspension-cultured calluses changed slowly, indicating that the calluses were gradually adapting to the environment. During the exponential stage (between days 6 and 12), the callus growth rate gradually increased and reached a maximum. The growth rate of the calluses gradually stabilized during the linear stage of days 12–24, and the change in the fresh and dry weights was linearly correlated with time. The deceleration stage lasted from days 24 to 36, during which the callus growth rate gradually decreased. At 36 days, the fresh and dry weights of the calluses reached their maximum. While they did not change significantly during the stationary stage of 36–39 days, these two weights began to decrease after 39 days.

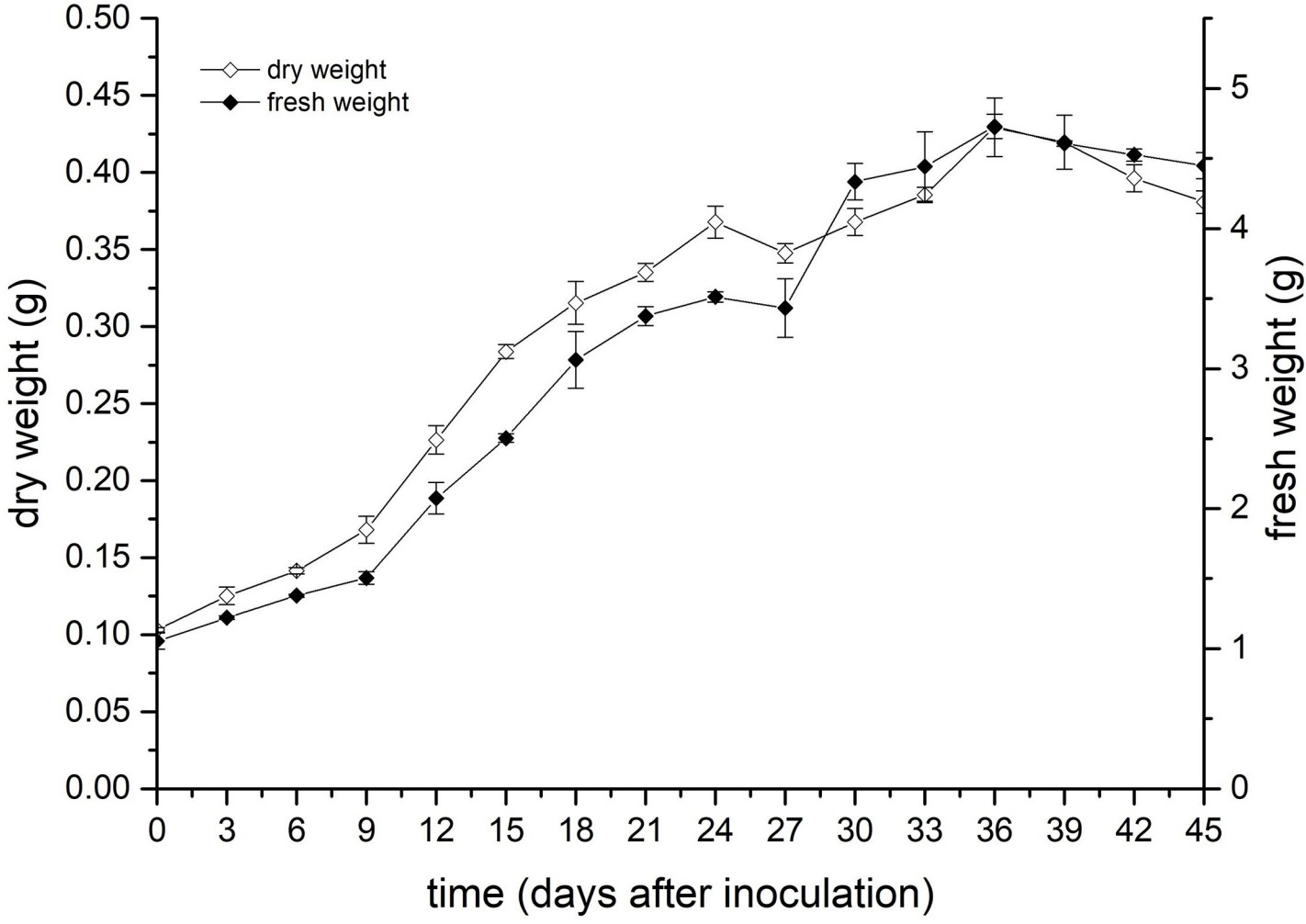

**Fig 1. Growth curves of the fresh and dry weights of suspension-cultured calluses during the whole culturing period.**

**Table 1. Mathematical modeling results of the growth curves of the fresh weight and dry weight of suspension-cultured calluses.**

| Trait | Mathematical model | Parameter | | | | |
|---|---|---|---|---|---|---|
| | | A1 | A2 | x0 | p/dx/ LOGx0 | $R^2$/F |
| Fresh weight | Logistic | 1.0540 | 4.7290 | 17.0845 | p = 3.0 | 0.9817 |
| | $f(x) = A2 + (A1-A2)/(1+ (x/x0)\hat{}p)$ | | | | | 721.56294 |
| | Boltzmann | 1.0540 | 4.7290 | 17.0845 | dx = 2.25 | 0.9231 |
| | $f(x) = A2 + (A1-A2)/(1 + exp((x-x0)/dx))$ | | | | | 168.85936 |
| | DoseResp | 1.0540 | 4.7290 | - | p = 0.11 | 0.9780 |
| | $f(x) = A1 + (A2-A1)/(1 + 10\hat{}((LOGx0-x)^*p))$ | | | | LOGx0 = 17.08 | 596.33032 |
| Dry weight | Logistic | 0.1032 | 0.4292 | 14.0785 | p = 3.0 | 0.9761 |
| | $f(x) = A2 + (A1-A2)/(1+ (x/x0)\hat{}p)$ | | | | | 613.94951 |
| | Boltzmann | 0.1032 | 0.4292 | 14.0785 | dx = 2.25 | 0.8878 |
| | $f(x) = A2 + (A1-A2)/(1 + exp((x-x0)/dx))$ | | | | | 128.45275 |
| | DoseResp | 0.1032 | 0.4292 | - | p = 0.11 | 0.9453 |
| | $f(x) = A1 + (A2-A1)/(1 + 10\hat{}((LOGx0-x)^*p))$ | | | | LOGx0 = 14.08 | 266.3732 |

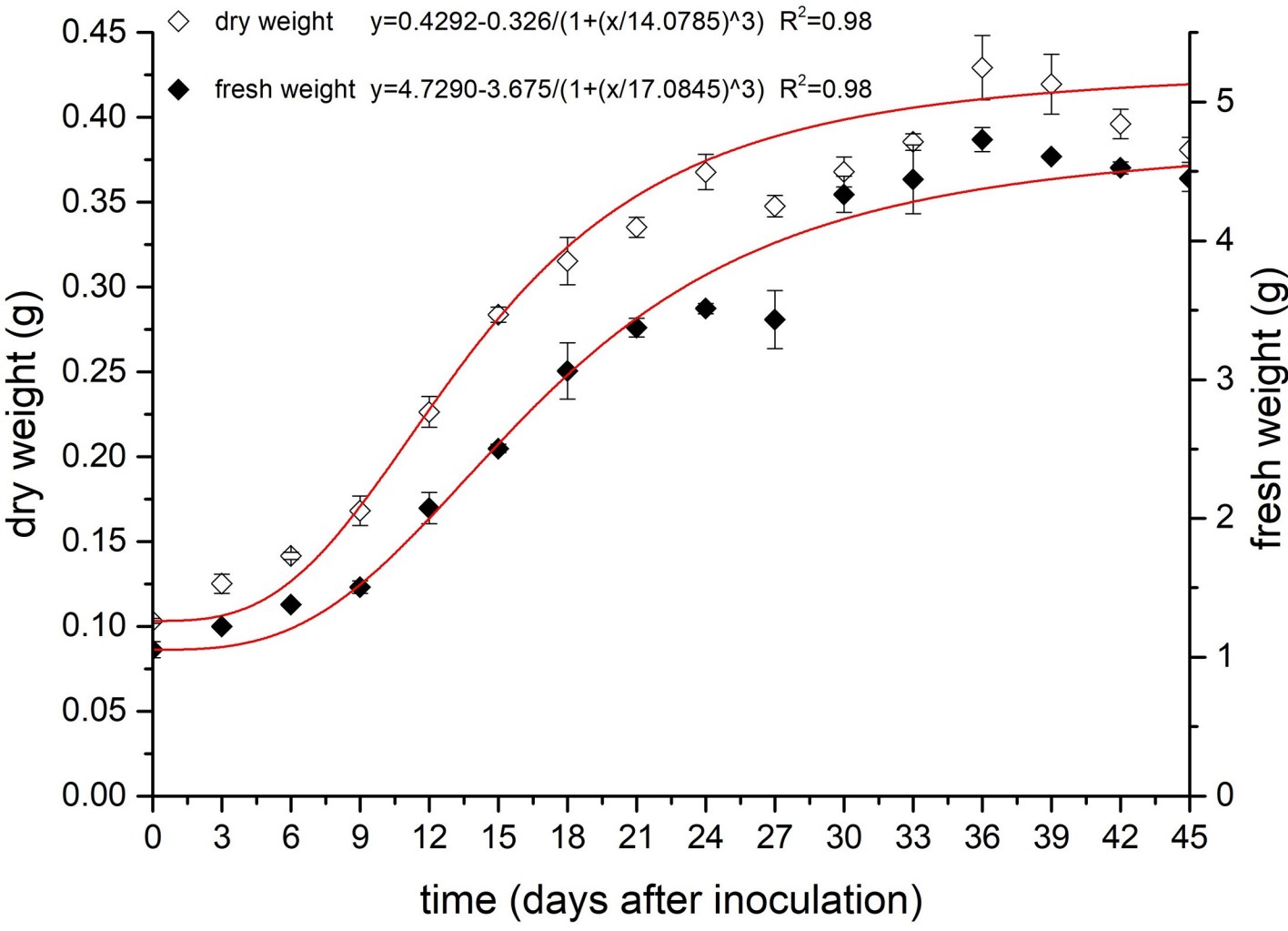

**Fig 2. Growth kinetic curves of fresh and dry weights of suspension-cultured calluses.**

**Callus growth curve function model analysis.** The Logistic function was presented as the cumulative amount of callus growth. To better understand the changes in suspension culture callus growth, we performed first-order derivatives (callus growth rate) and second-order derivatives (callus growth acceleration) of the simulated Logistic function and plotted the results of the derivation, as shown in Fig 3. It is reflected from the figure that the growth rate of the calluses was initially increasing. When it reached its peak, it was caused to decrease gradually by various factors affecting the growth rate. The overall trend first increased and then decreased. The extremums of the two curves showed that the growth acceleration of suspension-cultured calluses reached a maximum on day 7, and the callus growth rate reached a maximum on days 13 to 14.

## Measurement of secondary metabolite accumulation in suspension-cultured calluses

**Investigation of specificity.** The samples were chromatographically tested and compared with reference substances, and the filtrate of the suspension-cultured calluses and blank solution are shown in Fig 4. The results showed that the absorption peaks of the sample were

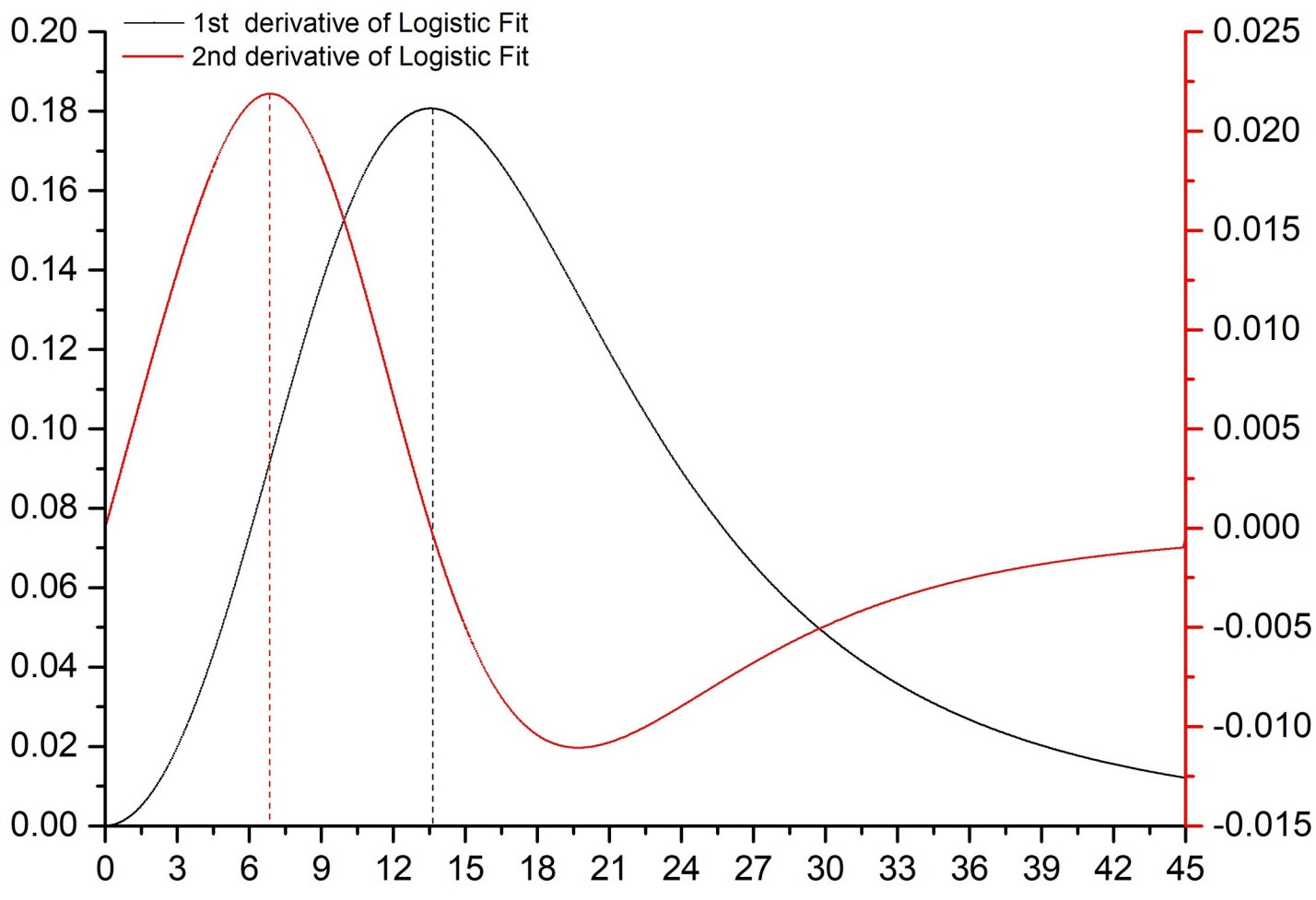

**Fig 3. Function graphs of the first derivative and the second derivative of the Logistic model.**

similar to those of the reference substances analyzed under the same conditions, and the blank solution indicated the absence of interferences. Moreover, there was no obvious detection of secondary metabolites in the callus culture medium, indicating that the samples accurately reflected the accumulated amount of secondary metabolites in the suspension-cultured calluses.

**Investigation of linear relationships.** The standard curves were drawn with the mass concentrations of *p*-hydroxybenzyl alcohol, dactylorhin A, militarine and coelonin as the abscissa (X) and the corresponding peak areas of the four secondary metabolites as the ordinate ($Y_{1-4}$) (S2 Fig). The regression equations were $Y_1 = 91387X-169.99$ ($R^2 = 0.9994$), $Y_2 = 36075X-712.50$ ($R^2 = 0.9996$), $Y_3 = 24341X-224.88$ ($R^2 = 0.9993$) and $Y_4 = 69896X-142.88$ ($R^2 = 0.9994$) for *p*-hydroxybenzyl alcohol, dactylorhin A, militarine and coelonin, respectively.

**Validation of the HPLC methodology.** The results of the precision test showed that the RSDs of the peak areas of *p*-hydroxybenzyl alcohol, dactylorhin A, militarine and coelonin were 0.83%, 0.40%, 1.36%, and 1.24%, respectively, while the RSDs of the retention times were 0.09%, 0.08%, 0.07%, and 0.16%, respectively (S2 Table). The results of the stability test showed that the RSDs of the peak areas of the four components were 0.71%, 1.73%, 1.86% and 2.80%, respectively (S3 Table), while the RSDs of the retention times were 0.86%, 0.50%, 0.76% and 1.61%, respectively, indicating that the solution had good stability within 12 hours. The results

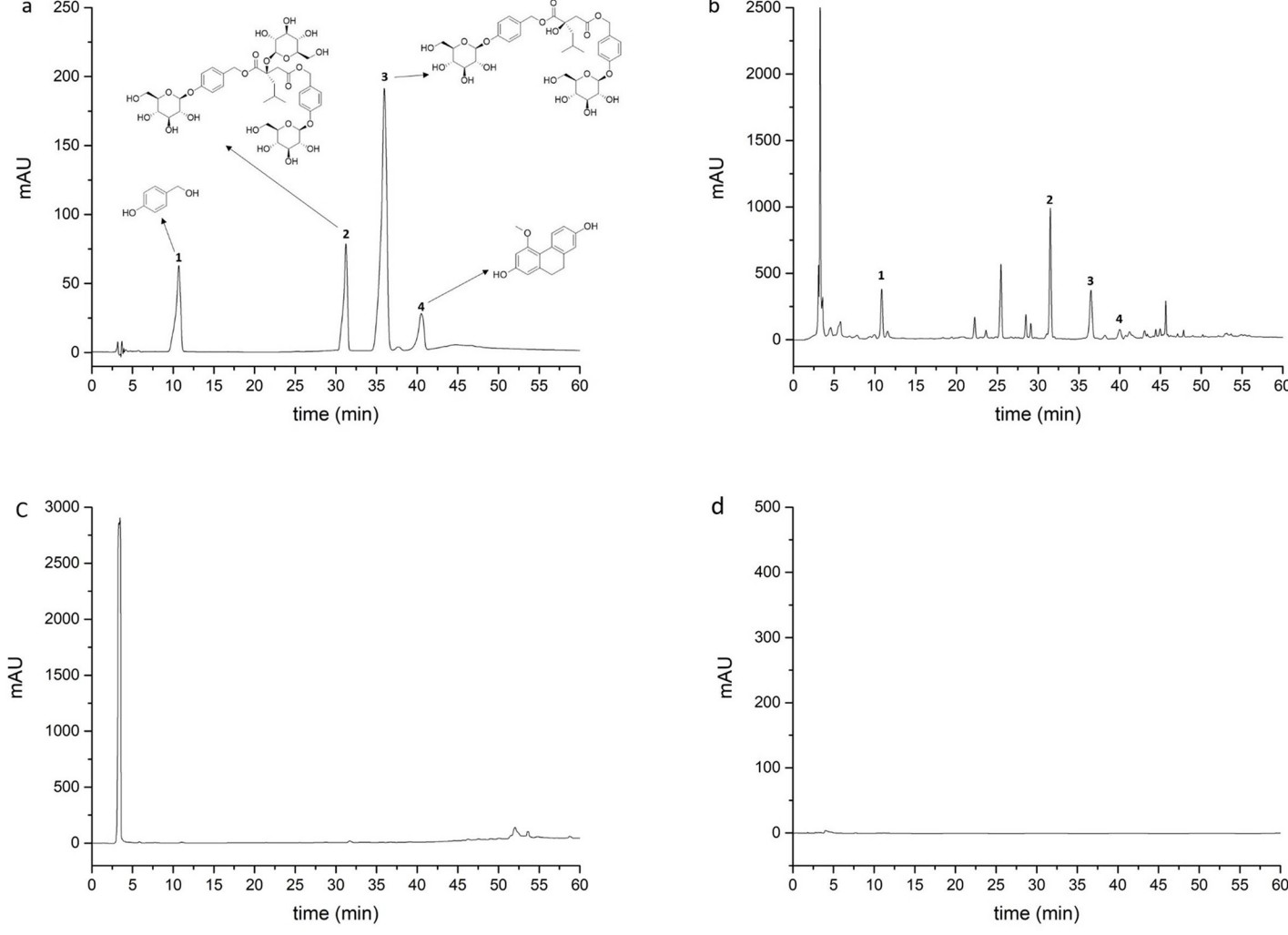

**Fig 4. HPLC chromatograms of various constituents.** a. Test samples. b. Reference substances. c. The filtrate of suspension-cultured calluses of *Bletilla striata*. d. Blank solution (70% methanol). 1. p-Hydroxybenzyl alcohol. 2. Dactylorhin A. 3. Militarine. 4. Coelonin.

of the repeatability experiment showed that the RSDs of the four components were 2.19%, 2.54%, 0.78% and 2.00%, respectively (S4 Table), which indicated good repeatability. The recoveries experiments of the four secondary metabolite results were 1.59%, 1.85%, 1.24% and 1.98% (S5 Table), showing that the method had good accuracy.

**Table 2. Mathematical model fitting results of the contents of four secondary metabolites.**

| Secondary metabolite | Mathematical model | Functional equation | $R^2$ |
|---|---|---|---|
| *p*-hydroxybenzyl alcohol | SGompertz | $f(x) = 1.0489^*\exp(-\exp(-0.0886^*(x-3.1379)))$ | 0.9121 |
| dactylorhin A | Exponential (*if* $x \leq 24$) | $f(x) = 5.7649 + 0.9840^*\exp(0.1084^*x)$ | 0.9015 |
| | ExpDec1 (*if* $x \geq 24$) | $f(x) = 365.0176^*\exp(-x/7.1996) +6.0012$ | 0.9568 |
| militarine | GaussAmp (*if* $x \leq 24$) | $f(x) = 2.2007+4.1171^*\exp(-0.5^*((x-24)/6.9654)^2)$ | 0.8443 |
| | Bradley (*if* $x \geq 24$) | $f(x) = -21.9151^*\ln(0.2391^*\ln(x))$ | 0.9715 |
| coelonin | BiDoseResp (*if* $x \leq 18$) | $f(x) = 0.1658+0.0833/(1+\text{pow}(10(2.5492-x)^*0.2778))+ 0.0833/(1+\text{pow}(10(13.2375-x)^*0.2778))$ | 0.9372 |
| | DoseResp (*if* $x \geq 18$) | $f(x) = 0.201 +0.1313/(1 + 10^\wedge((26.6223-x)^*-0.1852))$ | 0.9038 |

## Kinetic characteristics of secondary metabolite accumulation in the suspension system

The accumulation of the four secondary metabolites in calluses of different growth stages was calculated (S6 Table) and plotted as curves. The mathematical models were used to describe the changes in the accumulation of the four chemicals, as shown in Table 2. The results showed that the changes of secondary metabolites in suspension-cultured calluses were more complicated than callus growth. A single mathematical model could not sufficiently describe the measured data. Therefore, for the three secondary metabolites (dactylorhin A, militarine, coelonin), a multimathematical model and piecewise function were selected to describe the change in their cumulative amount, as shown in Fig 5.

According to the figure, the accumulation of *p*-hydroxybenzyl alcohol in suspension-cultured calluses was similar to callus growth, showing a gradual upward trend. The curve went smoothly, and the cumulant increased slowly over 24 days, and the accumulation reached a maximum at 39 days. There was a distinct maximum value in the cumulative curves of dactylorhin A, militarine and coelonin. The cumulative amount of dactylorhin A and militarine both reached a maximum at 24 days, and the cumulative amount of coelonin reached a maximum at 18 days.

## Discussion and conclusion

Compared with plant cultivation, suspension callus culturing has the advantages of a short growth cycle, easy separation of secondary metabolites and easy control of influencing conditions. Plant callus cultures makes it easier to obtain specific natural products from medicinal plants by inducing the synthesis of specific secondary metabolites. Therefore, using mathematical models to analyze callus growth and secondary metabolite synthesis and accumulation is important for revealing the synthesis mechanism of natural products such as secondary metabolic components, improving the yield of secondary metabolites and enhancing the medicinal value of medicinal plants[11]. In recent years, with the deepening of relevant research, the medicinal value of various secondary metabolites in *B. striata* has been confirmed by scientific research. For example, *p*-hydroxybenzyl alcohol can increase the expression of genes encoding antioxidant proteins after focal cerebral ischemia, which can deter oxidative stress and further damage to brain neurons[12]. Dactylorhin A and militarine can significantly improve memory impairment in mice that is caused by chemicals such as scopolamine, cycloheximide and alcohol[13], while 2,7-dihydroxy-4-methoxy-9,10-dihydrophenanthrene (coelonin) has certain antiviral activity as a kind of dihydrophenanthrene compound. Thus, it is meaningful to use plant callus liquid culture technology to regulate the synthesis of various secondary metabolites[14].

In this study, we simulated the growth of *B. striata* suspension-cultured calluses with a variety of mathematical function models and properly simplified the complex environment in which a group of calluses were grown with the mathematical model of growth kinetics. From the results of function fitting, all the mathematical models could accurately describe the growth of calluses at different growth stages and the changes of secondary metabolite accumulation. The growth of suspension-cultured calluses was divided into six stages: the lag stage, exponential stage, linear stage, deceleration stage, stationary stage and recession stage. Between days 13 and 14, the callus growth rate reached its maximum. After 39 days, callus growth did not show a significant decline in the functional model, but according to the actual data and observation of the state of the cultured calluses, we found that the cultured calluses showed obvious browning, indicating that the growth entered the recession stage. These phenomena were well presented in the functional model.

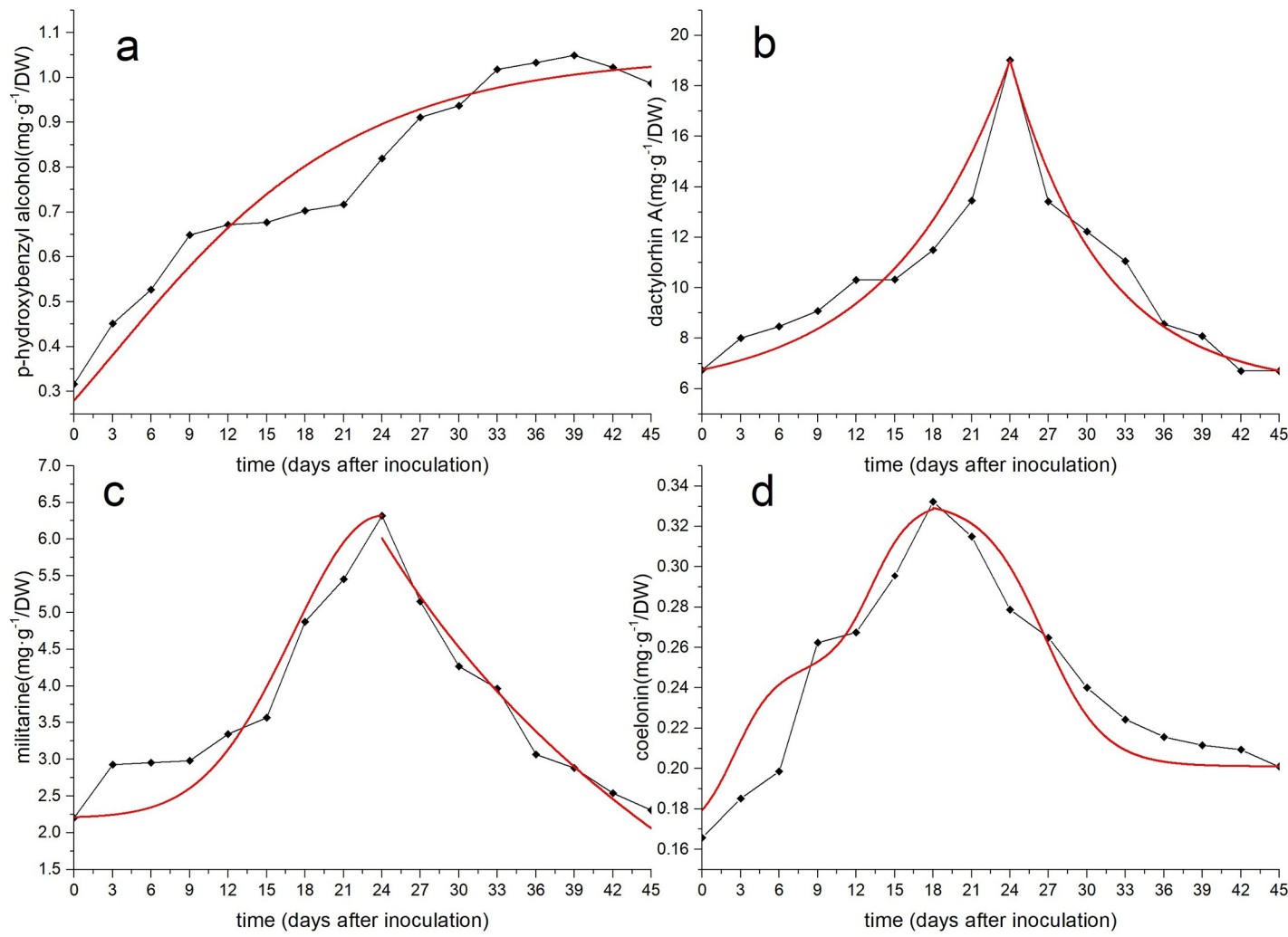

**Fig 5. Content change and fitting curves of four secondary metabolites.** a. p-Hydroxybenzyl alcohol. b. Dactylorhin A. c. Militarine. d. Coelonin.

However, the changes in the accumulation of secondary metabolites in the suspension-cultured calluses of *B. striata* were complicated and diversified. The change in the cumulative amount of *p*-hydroxybenzyl alcohol basically followed the change in the growth of *B. striata* suspension culture calluses, while the content of dactylorhin A, militarine and coelonin changed inconsistently with the growth of calluses. By comparing the growth curve with the curve of the accumulation of secondary metabolites, when the callus growth reached the end of the linear growth stage (days 12–24), the accumulation of secondary metabolites generally declined. At this stage, due to the reduction of nutrients in the liquid medium, the environmental inhibition began to be greater than the callus growth, and secondary metabolism, as a nonessential mechanism of plant life, gradually decreased. Dactylorhin A and militarine, as glycosidic compounds, are both secondary metabolites and energy storage substances that actively decompose to maintain life in adversity, which may result in a decrease in secondary metabolites in cells[15]. Fermentation kinetic studies on microorganisms have shown that *p*-hydroxybenzyl alcohol itself can act as an inducer for increasing extracellular polysaccharide production in fungi. In this study, we found that the curve of *p*-hydroxybenzyl alcohol accumulation gradually became smooth after 24 days, while dactylorhin A and militarine reached

maximum values at day 24. The effects of *p*-hydroxybenzyl alcohol on the key enzymes of polysaccharide synthesis in plant calluses and possible factors of synthesis promotion need to be further studied. It is well known that the production of secondary metabolites by plants is the result of their adaptation to their ecological environment during long-term evolution. The synthesis of these secondary metabolites is closely related to the growth state of plants and environmental conditions. Different types of secondary metabolites have different biosynthetic pathways at different stages of plant growth.

Further studies are needed to control the metabolic pathways, key enzymes and key regulatory genes so that we can improve the efficiency of suspension culture production of *B. striata* calluses and achieve efficiently direct induction of secondary metabolites by means of genetic engineering and metabolic engineering[16]. This study lays a theoretical and practical foundation for follow-up studies on the growth, mutant induction, secondary metabolite synthesis and bioreactor construction of *B. striata* calluses.

## Supporting information

**S1 Fig. Growth states of suspension-cultured calluses.** a. Suspension culture system. b. Calluses collected from the culture system after 3 days of incubation. c. Calluses after 9 days of incubation. d. Calluses collected from the culture system after 18 days of incubation. e. Calluses after 27 days of incubation. f. Calluses collected from the culture system after 33 days of incubation. g. Calluses collected from the culture system after 45 days of incubation.
(JPG)

**S2 Fig. Standard curves of four secondary metabolites.** a. 4-Hydroxybenzyl alcohol. b. Dactylorhin A. c. Militarine; d. Coelonin.
(JPG)

**S1 Table. Dry and fresh weights of suspension-cultured calluses during a 45-day culture period.**
(DOCX)

**S2 Table. Precision examination results of HPLC detection.**
(DOCX)

**S3 Table. Stability test results of HPLC detection.**
(DOCX)

**S4 Table. Repeatability test results of HPLC detection.**
(DOCX)

**S5 Table. Recovery test results of tested chemicals.**
(DOCX)

**S6 Table. Mean measurement of secondary metabolites during a 45-day culture period.**
(DOCX)

## Acknowledgments

This research was supported financially by the National Natural Science Foundation of China (31560079, 31560102, 31960074), the PhD Science Foundation of Zunyi Medical University (F-809), the Talent Growth Project of the Guizhou Education Department (KY[2017]194), the Research Project of the Guizhou Administration of Traditional Chinese Medicine (QZYY-2019-060), Science and Technology Department Foundation of Guizhou Province of China

(No. QKHPTRC[2019]5657), and Foundation of Excellent Young teachers of Zunyi Medical University of China (No.15zy-004). And we also thank a lot to Prof. Jishuang Chen for his kind inspiring to this research.

## Author Contributions

**Conceptualization:** Delin Xu.

**Data curation:** Shiji Xiao, Yanni ShangGuan.

**Formal analysis:** Shiji Xiao.

**Funding acquisition:** Delin Xu.

**Investigation:** Shebo Zhang.

**Methodology:** Houbo Liu.

**Project administration:** Lin Li, Delin Xu.

**Supervision:** Lin Li, Delin Xu.

**Writing – original draft:** Yinchi Pan, Zhongjie Chen, Delin Xu.

**Writing – review & editing:** Surendra Sarsaiya, Delin Xu.

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
