## [Decision Letter · Decision Letter 0]

19 Aug 2019

PONE-D-19-17799

Cell growth kinetics and accumulation of secondary metabolite of Bletilla striata Rchb.f. using cell suspension culture

PLOS ONE

Dear Dr. Li,

Thank you for submitting your manuscript to PLOS ONE. After careful consideration, we feel that it has merit but does not fully meet PLOS ONE’s publication criteria as it currently stands. Therefore, we invite you to submit a revised version of the manuscript that addresses the points raised during the review process.

We would appreciate receiving your revised manuscript by Oct 03 2019 11:59PM. To enhance the reproducibility of your results, we recommend that if applicable you deposit your laboratory protocols in protocols.io, where a protocol can be assigned its own identifier (DOI) such that it can be cited independently in the future. For instructions see: http://journals.plos.org/plosone/s/submission-guidelines#loc-laboratory-protocols

We look forward to receiving your revised manuscript.

Kind regards,

Jen-Tsung Chen, Ph.D.

Academic Editor

PLOS ONE

**Journal Requirements**

2. In your Methods section, please provide additional location information of the collection location, including geographic coordinates for the data set if available.

3. In your Methods section, please provide additional information regarding the permits you obtained for the work. Please ensure you have included the full name of the authority that approved the collection site access and, if no permits were required, a brief statement explaining why

**Comments to the Author**

1. Is the manuscript technically sound, and do the data support the conclusions?

Reviewer #1: Yes

Reviewer #2: Yes

2. Has the statistical analysis been performed appropriately and rigorously? 

Reviewer #1: Yes

Reviewer #2: Yes

3. Have the authors made all data underlying the findings in their manuscript fully available?

Reviewer #1: Yes

Reviewer #2: Yes

4. Is the manuscript presented in an intelligible fashion and written in standard English?

Reviewer #1: No

Reviewer #2: Yes

5. Review Comments to the Author

Reviewer #1: In this study, the authors cultivate suspension culture of Bletilla striata callus cells and recorded the changes in fresh weight, dry weight and secondary metabolite content during culture, which has certain significance for the sustainable use of Bletilla striata resources. However, I suggest that this manuscript needs to undergo major revision.

1. 1. Please add figures of Bletilla striata callus at different stages

2. In the abstract， "Bletilla striata is a well-recognized endangered medicinal plant due to inadequate natural reproduction with high market worth. "This statement is exaggerated, please describe it in a suitable vocabulary.

3. Does the dry weight of the sample affect the number of subsequent experimental samples?

4. English needs to be completely reviewed by a native speaker.

Reviewer #2: The work is well done and presented.

However, the material generated by the Authors did not respected the full description for a cell suspension (homogeneity or cellular individualization, for example), because this research with the initial steps of cellular individualization. It could be worth to mention this point.

p-hydroxybenzyl alcohol: p in italic.

This work is in my opinion could be acceptable for publication following these changes.

6. PLOS authors have the option to publish the peer review history of their article (what does this mean?). If published, this will include your full peer review and any attached files.

Reviewer #1: No

Reviewer #2: No

---

## [Author Response · Author response to Decision Letter 0]

12 Sep 2019

Response to Reviewers

Dear Editor and Reviewers:

Thanks for your letter and the comments on our manuscript “Cell growth kinetics and accumulation of secondary metabolite of Bletilla striata Rchb.f. using cell suspension culture”. 

We have learned much from the comments, which are fair, encouraging and also constructive. After careful consideration, corresponding addresses were made to our submitted manuscript. And the reviewer’s comments were responded point-point as fellows, which along with a clear indication of the location of the revised version. Hope these changes will make it fully meet PLOS ONE’s publication criteria. The main comments and our specific responses are detailed as below:

Reviewer #1: In this study, the authors cultivate suspension culture of Bletilla striata callus cells and recorded the changes in fresh weight, dry weight and secondary metabolite content during culture, which has certain significance for the sustainable use of Bletilla striata resources. However, I suggest that this manuscript needs to undergo major revision.

1. Please add figures of Bletilla striata callus at different stages.

Response: The figures of different stages of B. striata callus were added in Figure S1 (stated in Line 202).

2. In the abstract, "Bletilla striata is a well-recognized endangered medicinal plant due to inadequate natural reproduction with high market worth. "This statement is exaggerated, please describe it in a suitable vocabulary.

Response: According to your suggestion, we have revised this statement as “Bletilla striata is an endangered traditional Chinese medicinal plant with multiple uses and slow pace on regeneration of germplasm resource” in line 22 and 23 of the Revised manuscript.

3. Does the dry weight of the sample affect the number of subsequent experimental samples?

Response: As we explained in the article, the material used in the experiment was a uniform callus derived from B. striata seeds of the same genotype. Therefore, in the same batch of materials, although the number of subsequent experimental samples is limited due to the dry weight of the sample. However, the subsequent experimental materials can be repeated according to our experimental methods, and the same experimental results can be obtained.

4. English needs to be completely reviewed by a native speaker.

Response: According to your suggestion, this manuscript has been modified by a proferssor whose native language is English. And the English polishing tracks could be found in attached file of “Revised Manuscript with Track Changes”, which was edited by American Journal Experts (AJE) company with the supporting file of “Certification by AJE”.

Reviewer #2: 

1. The work is well done and presented. However, the material generated by the Authors did not respected the full description for a cell suspension (homogeneity or cellular individualization, for example), because this research with the initial steps of cellular individualization. It could be worth to mention this point.

Response: All the inoculated materials were derived from the uniform callus induced by the seeds in the mature capsules of the same artificial sister on the same single plant in Zheng’an County, Guizhou Province, China(28°56′N, 107°43′E). According to your suggestion, we have added this information in our Methods section of the Revised manuscript from Line 92 to Line 94.

2. p-hydroxybenzyl alcohol: p in italic. This work is in my opinion could be acceptable for publication following these changes.

Response: The spelling mistake of p-hydroxybenzyl alcohol has been corrected in Line 41, 84, 111, 161, 164, 174, 180, 257, 262, 266, 288, 307, 331, 343, 345, 347 and 476.

---

## [Decision Letter · Decision Letter 1]

7 Oct 2019

Callus growth kinetics and accumulation of secondary metabolites of Bletilla striata Rchb.f. using a callus suspension culture

PONE-D-19-17799R1

Dear Dr. Li,

We are pleased to inform you that your manuscript has been judged scientifically suitable for publication and will be formally accepted for publication once it complies with all outstanding technical requirements.

With kind regards,

Jen-Tsung Chen, Ph.D.

Academic Editor

PLOS ONE

Additional Editor Comments (optional):

Reviewers' comments:

Reviewer's Responses to Questions

**Comments to the Author**

1. If the authors have adequately addressed your comments raised in a previous round of review and you feel that this manuscript is now acceptable for publication, you may indicate that here to bypass the “Comments to the Author” section, enter your conflict of interest statement in the “Confidential to Editor” section, and submit your "Accept" recommendation.

Reviewer #1: All comments have been addressed

Reviewer #2: All comments have been addressed

2. Is the manuscript technically sound, and do the data support the conclusions?

Reviewer #1: Yes

Reviewer #2: Yes

3. Has the statistical analysis been performed appropriately and rigorously? 

Reviewer #1: Yes

Reviewer #2: Yes

4. Have the authors made all data underlying the findings in their manuscript fully available?

Reviewer #1: Yes

Reviewer #2: Yes

5. Is the manuscript presented in an intelligible fashion and written in standard English?

Reviewer #1: Yes

Reviewer #2: Yes

6. Review Comments to the Author

Reviewer #1: I’m appreciated that authors have done the recommended revisions accordingly and replied to several comments. So I recommend this article to be published.

Reviewer #2: (No Response)

7. PLOS authors have the option to publish the peer review history of their article (what does this mean?). If published, this will include your full peer review and any attached files.

Reviewer #1: No

Reviewer #2: No

---

## [Editor Report · Acceptance letter]

11 Feb 2020

PONE-D-19-17799R1 

Callus growth kinetics and accumulation of secondary metabolites of *Bletilla striata* Rchb.f. using a callus suspension culture 

Dear Dr. Li:

I am pleased to inform you that your manuscript has been deemed suitable for publication in PLOS ONE. Congratulations! Your manuscript is now with our production department. 

With kind regards,

on behalf of

Dr. Jen-Tsung Chen 

Academic Editor

PLOS ONE